# Factors Associated with Hospitalized Community-Acquired Pneumonia among Elderly Patients Receiving Home-Based Care

**DOI:** 10.3390/healthcare12040443

**Published:** 2024-02-08

**Authors:** Jui-Kun Chiang, Hsueh-Hsin Kao, Yee-Hsin Kao

**Affiliations:** 1Department of Family Medicine, Dalin Tzu Chi Hospital, Buddhist Tzu Chi Medical Foundation, No. 2, Minsheng Road, Dalin, Chiayi 622, Taiwan; jkch68@gmail.com; 2Department of Radiation Oncology, Taichung Veterans General Hospital, Taichung 40201, Taiwan; 3Department of Family Medicine, Tainan Municipal Hospital (Managed by Show Chwan Medical Care Corporation), 670 Chung-Te Road, Tainan 701, Taiwan

**Keywords:** community-acquired pneumonia (CAP), home-based care, elderly

## Abstract

(1) Background: Pneumonia stands as a prevalent infectious disease globally, contributing significantly to mortality and morbidity rates. In Taiwan, pneumonia ranks as the third leading cause of death, particularly affecting the elderly population (92%). This study aims to investigate factors associated with community-acquired pneumonia (CAP) among elderly individuals receiving home-based care. (2) Methods: Conducted between January 2018 and December 2019, this retrospective study involved a medical chart review of elderly patients under home-based care. A multiple logistic regression model was employed to identify factors associated with CAP in this demographic. (3) Results: Analysis encompassed 220 elderly patients with an average age of 82.0 ± 1.1 years. Eighty-five patients (38.6%) were hospitalized for CAP. Predominant diagnoses included cancer (32.3%), stroke (24.5%), and dementia (23.6%). Significant predictors of CAP for elderly patients under home-based care included male gender (odds ratio [OR] = 4.10, 95% confidence interval [CI]: 1.95–8.60, *p* < 0.001), presence of a nasogastric (NG) tube (OR = 8.85, 95% CI: 3.64–21.56, *p* < 0.001), and a borderline negative association with the use of proton pump inhibitors (PPIs) (OR = 0.37, 95% CI: 0.13–1.02, *p* = 0.0546). End-of-life care indicators for these patients with CAP included an increased number of hospital admission days in the last month of life (OR = 1.13, 95% CI: 1.08–1.18, *p* < 0.001) and a higher likelihood of hospital death (OR = 3.59, 95% CI: 1.51–8.55, *p* = 0.004). (4) Conclusions: In the current study, significant predictors of CAP among elderly patients receiving home-based care included the presence of an NG tube and male gender, while the use of PPIs was borderline inversely associated with the risk of CAP. Notably, more admission days in the last month of life and death in the hospital were found to be associated with end-of-life care for this group.

## 1. Introduction

Globally, the rapidly aging population poses a significant challenge for future provision of end-of-life care [1]. This demographic shift is expected to lead to an increased demand for various forms of healthcare services, including primary health care, home care, long-term care, and end-of-life care [2]. In 2018, Taiwan met the World Health Organization’s definition of an aged population, with over 14% of its residents aged 65 years and older. In 2019, there were a total of 175,424 deaths in Taiwan, with the following distribution: 91,713 (52.3%) in hospitals, 56,529 (32.2%) at home, 21,091 (12.0%) at other locations, and 6091 (3.5%) in nursing facilities [3]. Notably, pneumonia was the third leading cause of death in 2019, accounting for about 15,185 (8.7%) deaths, and a substantial majority of these cases, 13,936 (91.8%), were among the elderly [4]. The increasing aging population and a higher prevalence of hospital deaths have prompted reforms in the healthcare systems and policies. “Aging and dying in place” has become the guiding principle for addressing the challenges of rapid aging and providing care for elderly patients in Taiwan. The term “in place” refers to care provided outside of the hospital and encompasses both staying at home or in nursing facilities. To support this approach, Taiwan’s National Health Insurance Administration initiated the “Integrated Home Care” pilot project for home care and introduced new policies for long-term care, all aimed at promoting the concept of “aging and dying in place” to effectively address the challenges associated with an aging population and the desire for end-of-life care [5].

The demand for home healthcare services is steadily increasing due to the growing preference of elderly individuals to age in their own homes, a trend driven by the global aging population. A previous cohort study conducted in Taiwan revealed that in 2004, only 9 out of every 1000 elderly patients received home care services. This rate was notably lower than the prevalence of disabled elderly individuals in Taiwan. Furthermore, the study observed a higher utilization of home care services by females compared to males. The majority of skilled nursing services provided were related to nasogastric (NG) tube replacements [6]. Home-based care offers several advantages, including helping to manage healthcare facility capacity constraints, reducing the risk of hospital-acquired infections, enhancing the continuity of care, and lowering the likelihood of hospital readmissions and mortality [7].

Pneumonia in the elderly can progress rapidly, with a poor prognosis, making them particularly vulnerable to severe pneumonia. The mortality rate for severe pneumonia can be as high as 20% [8]. Previous studies have reported that elderly patients who are bedridden and fed through tubes face a significantly higher risk of mortality due to pneumonia [9,10]. The mortality rate from aspiration pneumonia is largely influenced by the volume and content of aspirate, and it can range as high as 70% [11,12].

Understanding the risk factors for community-acquired pneumonia (CAP), particularly among elderly patients, is of utmost importance. Elderly individuals often have weakened immune systems and may suffer from various underlying health conditions, making them more susceptible to CAP and its complications. Identifying and managing these risk factors can assist healthcare providers and public health officials in taking proactive measures to reduce the incidence of CAP in this high-risk population and enhance their overall health and well-being. The main objective of this study is to investigate the factors related to community-acquired pneumonia (CAP) in elderly patients undergoing home-based care. The focus is on elderly individuals with advanced illnesses who receive care in their homes. Given the elevated mortality associated with CAP, the in-hospital mortality rate is recorded at 11%. Furthermore, those who are discharged experience an additional mortality rate of 33.6% in the subsequent year [13]. It is noteworthy that there is a relative dearth of research in the existing literature concerning end-of-life care for elderly patients with advanced illnesses, specifically those with CAP, who are receiving home-based care. In this study, we have included several variables related to end-of-life quality indicators, which have been adapted from those originally designed for patients with advanced cancer, with necessary modifications [14]. Additionally, the current study investigates the relationship between end-of-life care indicators for elderly patients receiving home-based care and the occurrence of CAP.

## 2. Materials and Methods

### 2.1. Ethical Considerations

We conducted a retrospective review of deceased patients with advanced illnesses who had received home-based care in a secondary teaching hospital in Taiwan between January 2018 and December 2019. This study received approval from the Research Ethics Committee of the institutional review board of the Tainan Municipal Hospital (managed by Show Chwan Medical Care Corporation, Tainan, Taiwan), Taiwan (SCMH_IRB No: 1090104).

### 2.2. Materials

#### 2.2.1. Data Collection and Definition of Variables

For descriptive purposes, the primary diagnoses were categorized using International Classification of Diseases, Tenth Revision, Clinical Modification: cancer (C00–C97), COPD (J43 and J44), CHF (I50 and I50.x), ESRD on hemodialysis (Z99.2), dementia (F00, F00.x, F01, F01.x, F02, F02.x, and F03), parkinsonism (G20.x, G21.x), cirrhosis (K74.0, K74.60, and K74.69), and cerebrovascular accidents (stroke; I60–I68.x). The following information was extracted from the medical records: gender, age, primary diagnosis (including cancer, chronic obstructive pulmonary disease [COPD], dementia, parkinsonism, stroke, cirrhosis, end-stage renal disease [ESRD], and congestive heart failure [CHF]), the presence of invasive medical devices (such as nasogastric [NG] tubes, urinary catheters, and tracheostomy tubes), and the recorded place of death as per the death certificate. Additionally, clinical signs and symptoms observed during the initial home visit of patients were documented. The data obtained during the initial home visit in this study were adapted from Lee et al. [15]. Body temperature was recorded, and the presence of a fever episode was defined as a core temperature of >38 °C. An experienced registered nurse collected data on demographics, clinical symptoms and signs, laboratory test results, admission days in the final month of life, and the eventual places of death. The accuracy of the data was subsequently verified by one of the authors. 

#### 2.2.2. Definition of Variables

In the current study, we defined patients with pneumonia as those who met the following criteria: having a primary diagnosis recorded on both the admission and discharge medical records with ICD-10-CM codes, specifically J11.0, J12, J13, J14, J15, J16, J17, and J18; exhibiting at least one clinical symptom or sign (such as productive cough, fever with a body temperature > 38 °C, dyspnea, chest pain, crackles on auscultation, or a change in consciousness); and displaying a chest radiograph showing an opacity consistent with the presence of pneumonia [16,17]. The use of the NG tube and urinary catheter was defined as their timing and duration being for more than 1 week before the diagnosis of CAP. In the current study, patients with NG tube/urinary catheter duration exceeding 7 days received a positive code, while others received a negative code. Additionally, we investigated specific medications that may be associated with CAP [18,19,20]. The medications under investigation included gastroprokinetic agents (such as Metoclopramide, Mosapride, and Domperidone), as well as acid-suppressive agents like proton-pump inhibitors (PPIs) and histamine type-2 receptor antagonists (H_2_ blockers). Gastroprokinetic agents are medications that enhance gastrointestinal motility by increasing the frequency or strength of contractions. Previous studies have reported that the use of gastroprokinetic agents reduces the incidence of aspiration pneumonia by promoting upper gastrointestinal motility and preventing gastroesophageal reflux [21]. The mechanism underlying PPI-associated pneumonia is believed to be multifactorial, primarily resulting from the compromised stomach’s ‘acid mantle’, which normally serves as a protective barrier against the colonization of acid-labile pathogenic bacteria. Other acid-suppressive agents, such as H_2_ blockers, decrease gastric acid secretion by parietal cells. With the administration of these acid-suppressive agents, bacteria may encounter lower gastric acid levels, potentially leading to their aspiration into the respiratory tract and an increased risk of developing CAP [18,20]. The period of medication exposure was calculated from the date of CAP diagnosis, and patients who had received these medications for more than 1 week were included in the study.

The end-of-life (EOL) quality indicators, originally derived from Earle’s study and initially applied to cancer patients, underwent modifications and were assessed in the present study [14,22]. Regarding hospice palliative care in Taiwan, the team typically consists of specialized physicians, trained nurses, social workers, qualified chaplains, and qualified volunteers. In this study, we scrutinized end-of-life (EOL) care indicators, which encompassed an increased count of hospital admission days during the last month of life, the receipt of intensive care unit (ICU) services in the final month of life, the lack of hospice palliative care consultation, and death in the hospital. 

### 2.3. Method

#### 2.3.1. Data Design and Setting

The inclusion criteria for the current study were as follows: age ≥ 65 years, the presence of an advanced illness, a duration of home-based care exceeding 1 month, and patients who died between 2018 and 2019. In Taiwan, patients are eligible to apply for home-based care if they meet the following three criteria: a limited performance status, such as being bedridden or using a wheelchair for more than 50% of their waking hours; a clear need for medical or nursing care; and the presence of chronic conditions that require long-term or continuous nursing care following hospital discharge, combined with an inability to travel for medical treatment [23]. For the study period, if a patient had been hospitalized for CAP more than once, only data from the first admission were included. The exclusion criterion was patients discharged from an acute care hospital within 14 days. Participants were categorized into two groups: the first group comprised elderly patients with advanced illnesses who received home-based care and were hospitalized due to CAP (referred to as the “P group”), and the second group included those without CAP (referred to as the “non-P group”). In the current study, all patients received home-based care. In the event of a patient developing CAP, their family could seek assistance from the home care health team for management. If the patient’s condition worsened or if the family was unable to provide care at home, the patient would be transported back to the hospital for admission. Another reason for patients with CAP being sent back to the hospital was the availability of hospitalization facilities in Taiwan.

#### 2.3.2. Study Outcome

As previously mentioned, variables from the medical records were collected to explore the factors associated with CAP among elderly patients receiving home-based care.

### 2.4. Statistical Analysis

All statistical analyses were performed on R (version 4.2.3; R Foundation for Statistical Computing, Vienna, Austria). A two-sided *p* of ≤0.05 was considered statistically significant. The distribution properties of continuous variables were expressed using means ± standard deviations and categorical variables using frequencies and percentages. Normality was examined using the Shapiro–Wilk test. For the univariate analysis, the two-sample t test, Wilcoxon rank-sum test, chi-square test, and Fisher’s exact test were used to examine differences in the distributions of continuous and categorical variables between the two groups as indicated.

Multivariate analysis was conducted by fitting multiple logistic regression models with the stepwise variable selection procedure to explore the associated factors. In logistic regression, continuous predictors (independent variables) are assumed to have a linear relationship with the log odds of the outcome, known as the “linearity of the logit” assumption. The Hosmer–Lemeshow test is commonly employed to assess the goodness of fit in a logistic model, while the Nagelkerke R-squared quantifies the proportion of the total variation in the dependent variable explained by independent variables in the current model. Furthermore, we assessed the goodness of fit of the final logistic regression model based on the estimated area under the receiver operating characteristic curve (AUC; also called the “c statistic”). Statistical tools for regression diagnostics, including checking multicollinearity, were applied to ascertain any problems associated with the regression model or data.

## 3. Results

In the current study, a total of 228 elderly patients who were receiving home-based care and passed away between 2018 and 2019 were initially included. After excluding patients with missing data (*n* = 8), a total of 220 elderly patients were included for analysis. Participants were categorized into two groups: elderly patients with advanced illnesses who received home-based care and were hospitalized due to CAP (the “P group”) and those without CAP (the “non-P group”). The study design is depicted in Figure 1. The most common primary diagnosis among the patients was cancer (71 cases, 32.3%), followed by stroke (54 cases, 24.5%), and dementia (52 cases, 23.6%). The prevalence of CAP was significantly lower among individuals with cancer compared to those without cancer (*p* = 0.017). The predominant clinical symptoms/signs were unconsciousness, followed by fever and dyspnea. Out of the 220 patients, 67 (30.5%) received H_2_ blockers, 53 (24.1%) received gastroprokinetic agents, and 32 (14.5%) received PPIs.

Regarding end-of-life care indicators for elderly patients receiving home-based care, the mean number of days spent in the hospital during the last month of life was 8.4 ± 9.2 days. Patients in the P group had a significantly longer duration of hospital admission (13.6 ± 9.2 days) compared to those in the non-P group (5.1 ± 7.6 days). Among the 147 patients (66.8%) who died, those in the P group had a significantly higher likelihood of dying in a hospital (84.7%) compared to those in the non-P group (55.6%) (Table 1). In the Pearson correlation analysis, CAP is significantly associated with the ICU (*p* = 0.015, *r* = 0.826), while it does not show a significant association with other predictors of poor-quality end-of-life (EOL) care in our study.

In the univariate analysis, several factors were found to be associated with a higher likelihood of developing hospitalized CAP. These factors included male patients (odds ratio [OR] = 2.33, 95% confidence interval [CI]: 1.34–4.08, *p* = 0.003), patients with parkinsonism (OR = 12.10, 95% CI: 1.45–99.60, *p* = 0.021), and patients with an NG tube (OR = 4.57, 95% CI: 2.31–9.02, *p* < 0.001), with the exception of diastolic blood pressure (OR = 0.97, 95% CI: 0.95–0.99, *p* = 0.016) (Table 2).

End-of-life care indicators associated with patients in the P-group (those who developed hospitalized CAP) included having signed a Do Not Resuscitate (DNR) order (OR = 1.96, 95% CI: 1.01–3.84, *p* = 0.048), a higher number of admission days in the last month of life (OR = 1.12, 95% CI: 1.09–1.16, *p* < 0.001), and dying in the hospital (OR = 4.43, 95% CI: 2.24–8.76, *p* < 0.001).

By multiple logistic regression analyses, the factors associated with hospitalized CAP for elderly home-based patients were male (OR = 4.10, 95% CI: 1.95–8.60, *p* < 0.001) and presence of NG tube (OR = 8.85, 95% CI: 3.64–21.56, *p* < 0.001). Additionally, the use of PPI medications showed a borderline negative association (OR = 0.37, 95% CI: 0.13–1.02, *p* = 0.0546). The indicators of end-of-life care associated with patients in the P-group were more admission days in the last month of life (OR = 1.13, 95% CI: 1.08–1.18, *p* < 0.001), and death in the hospital (OR = 3.59, 95% CI: 1.51–8.55, *p* = 0.004) (Table 3). The Hosmer–Lemeshow test passed successfully (*p* = 0.11). The Nagelkerke R-squared value stands at 0.485. Furthermore, the area under the ROC curve is 0.859, with a 95% confidence interval of 0.808 to 0.910, signifying strong discriminatory performance by the model. We categorized the admission days in the last month into more than 7 days or not, instead of using continuous admission days. We found that this categorization remains a significant factor in the final model (*p* < 0.001).

## 4. Discussion

In the current study, several predictors of hospitalized CAP in elderly patients receiving home-based care were identified. These predictors included male gender and the presence of an NG tube. Conversely, the use of PPIs was borderline inversely associated with CAP. Additionally, the indicators of end-of-life care associated with elderly patients with hospitalized CAP receiving home-based care included a higher number of admission days in the last month of life and death in the hospital.

Community-acquired pneumonia (CAP) is the second most common cause of hospitalization and the leading infectious cause of death in the United States [24]. Furthermore, nearly 9 percent of patients initially hospitalized with CAP will experience rehospitalization due to a new episode of CAP within the same year. In Taiwan, the standardized mortality rate for pneumonia was 30.0 per 100,000 population every year in 2019. Pneumonia held the position of the third leading cause of death in Taiwan. Notably, the majority of pneumonia-related deaths in Taiwan occurred among elderly patients, constituting 92% of the cases. Understanding the factors associated with pneumonia in elderly patients receiving home-based care is crucial. A systematic review study reported that risk factors for CAP in adults included age, smoking, environmental exposures, malnutrition, previous CAP, chronic bronchitis/chronic obstructive pulmonary disease, asthma, functional impairment, poor dental health, immunosuppressive therapy, oral steroids, and treatment with gastric acid-suppressive drugs [25]. Another study reported a significant association between hospitalized CAP in home health care patients and male gender as well as patients with NG tubes [26]. A previous review study reported that the incidence of CAP was higher in males and increased with age in both genders [27,28]. Several contributing factors were identified, including unhealthy habits, chronic lung diseases, and medication use [29]. In the current study, we found that male elderly patients receiving home-based care had a higher risk of CAP.

Nasogastric (NG) tube placement is a valuable method for maintaining enteral access to provide nutrition, hydration, and medication to patients with dysphagia resulting from conditions such as stroke, brain injury, neurodegenerative diseases, or obstruction due to cancer. A previous clinical trial reported that early tube feeding would significantly improve survival after a stroke, albeit with a poor functional outcome [30]. Another prior review study indicated that the use of NG tubes improved hospital survival compared to other procedures. Nevertheless, there is insufficient high-quality evidence to support the improvement of health status and quality of life with the use of NG tubes in patients receiving palliative care [31]. The decision to initiate NG tube feeding for individuals experiencing dysphagia or unconsciousness might pose a challenge for their family. The Patient Right to Autonomy Act, passed in Taiwan in 2019, serves the legislative purpose of respecting patients’ medical autonomy, safeguarding their rights to a dignified end-of-life, ensuring everyone’s right to make informed choices, and the acceptance or refusal of medical treatment, including tube feeding (e.g., NG tube). The act aims to guarantee legal protection and implementation of patients’ end-of-life wishes, even when they are unconscious or unable to clearly express their intentions. However, it is worth noting that previous systematic review studies have reported concerns associated with tube feeding (including NG and PEG tubes). These concerns include an increased mortality rate and potential tube-related complications, such as aspiration pneumonia. Prolonging survival days and improving the nutritional status for hospitalized patients with advanced dementia were not consistently observed with tube feeding [32]. Another previous systematic review study found no significant difference in mortality or adverse events, including aspiration pneumonia, between the NG tube and PEG tube feeding groups [33]. Additionally, previous studies have indicated that a high gastric residue is associated with an increased risk of aspiration in tube-fed patients [34]. However, it is important to note that continuous pump feeding may not necessarily be superior to bolus feeding. Therefore, it is recommended to consider increasing the frequency of feeding while reducing the volume of each feeding to mitigate the risk [26]. In response to these concerns, a “No NG Tube for Life” policy has been in effect since April 2022. The primary objective of this policy is to encourage the removal of nasogastric tubes in patients. It includes incentives provided by Taiwan’s National Health Insurance to promote the removal of long-term indwelling nasogastric tubes [3]. In summary, preventing pneumonia in elderly, tube-fed male patients is a significant concern and requires attention and education for the staff providing home-based care.

Prolonged use of NG tubes can indeed increase the risks of esophageal reflux, irritation, ulceration, and bleeding [35]. In such cases, treatment often involves the administration of acid-suppressive medications. Proton pump inhibitors (PPIs) have become widely used in the treatment of peptic ulcer disease, Helicobacter Pylori infection, gastroesophageal reflux disorder, and other hypersecretory conditions. PPIs and H_2_ blockers are typically prescribed to patients with upper gastrointestinal bleeding, peptic ulcers, and acid reflux. In the current study, 32 (14.5%) patients were prescribed PPIs, and 67 (30.5%) were prescribed H_2_ blockers. A nationwide population-based study reported a 73% increased risk of pneumonia associated with the use of PPIs [18]. Another study found that the risk associated with PPI use was higher during the first 1-30 days of treatment compared to longer durations exceeding 30 days [18,36]. In the current study, it was noted that the utilization of PPI medications appeared to be a borderline negative predictor (*p* = 0.0546). It is crucial to emphasize that this *p*-value does not reach the conventional threshold for statistical significance, typically set at 0.05. 

Mortality rates due to pneumonia showed a significant increase with age. In-hospital, 30-day, and one-year mortality rates were twice as high among older adults aged ≥ 60 years (19.8%, 24.5%, and 47.4%) compared to their younger counterparts (8.4%, 10.0%, and 21.3%) [37]. Moreover, in the present study, we included 71 patients with cancer, constituting 32.3% of the total patient population. The remaining patients also had advanced illnesses and were receiving home-based care. Our additional objective was to assess the quality of end-of-life care provided to elderly patients with advanced illnesses receiving home-based care. To achieve this, we included several variables related to end-of-life quality indicators, which were originally designed for patients with advanced cancer [14]. In cases where the patient had an advanced illness, and CAP was identified as the primary event leading to the patient’s death, we considered CAP as a contributing factor. Assessing the quality of end-of-life care for elderly patients with advanced illnesses receiving home-based care is crucial for healthcare providers, particularly when these patients experience CAP.

Prolonged hospital stays can impose a significant burden on patients and their families, with admission days exceeding 14 days in the last month of life being considered a poor-quality indicator of end-of-life (EOL) care [22]. A cohort study conducted in Taiwan found that elderly patients with advanced liver cancer had an increased number of days spent in hospital during the last month of their lives [38]. In the current study, we observed that the mean duration of hospital admission in the last month was 8.4 ± 9.2 days for the entire study population. However, for the group of elderly patients with CAP, this duration was notably longer at 13.6 ± 9.2 days. These findings suggest that elderly patients with advanced illnesses who received home-based care and experienced hospitalization due to CAP had a higher number of hospital admission days during the last month of their lives. It appears that CAP may contribute to an extension of the length of hospital stays in the final month of life. This prolonged duration of hospital admissions can likely be attributed to the increased morbidity and mortality rates associated with CAP in elderly patients. A previous study has also reported a correlation between longer hospital stays and a higher 30-day mortality rate in elderly patients with CAP [39].

Cancer patients who passed away in a hospital setting often experienced increased physical and emotional distress, leading to a reduced quality of life [40]. Hospital deaths are considered an unfavorable indicator of the quality of end-of-life care for individuals with cancer [22]. However, for patients without cancer, the occurrence of death in a hospital should not automatically be seen as an indicator of substandard care, as their treatment goals may not have been solely focused on survival. A cohort study reported that the percentage of hospital deaths for advanced cancer patients was 65.4% [41]. In the current study, a total of 147 patients (66.8%) passed away in the hospital, with a higher proportion of them belonging to the CAP group (72 patients, 84.7%). One possible reason for this could be that these patients required hospital-based medical care until the time of their death due to the severity of their condition and the need for specialized treatment.

In the present study, we observed that 12 (5.5%) elderly patients with advanced illnesses receiving home-based care were admitted to the Intensive Care Unit (ICU). Among these, five (5.9%) patients belonged to the P group, and seven (5.2%) were in the non-P group. Our statistical analysis also indicated no significant differences between these two groups. Additionally, our study revealed that 165 (75%) of the patients had signed a Do Not Resuscitate (DNR) order. Among these, 70 (82.4%) patients belonged to the P group, and 95 (70.4%) were in the non-P group. Our statistical analysis did not reveal any significant differences between the two groups.

In the current study, towards the end of life, 81 (36.8%) elderly patients with advanced illnesses who received home-based care had received hospice palliative care. Among those with CAP, 28 (32.9%) received hospice care, while those without CAP amounted to 53 (39.3%) who received hospice palliative care. To enhance the quality of end-of-life care for elderly patients with advanced illnesses who receive home-based care and are hospitalized due to CAP, we recommend the implementation of hospice and palliative care for this specific population. The scope of hospice and palliative care has been expanded to encompass other serious illnesses, such as end-stage renal disease (ESRD), advanced dementia, congestive heart failure (CHF), and chronic obstructive pulmonary disease (COPD). The current study has several limitations. Firstly, being a retrospective study, it relied on medical record reviews, which come with inherent constraints typical of this study design, such as missing data, selection biases, and recall biases. Patients may have utilized healthcare services from other sources, and records from outside our hospital were unavailable for analysis, potentially leading to incomplete data. Secondly, data omissions are common in retrospective studies, and we did not include data from eight cases (3.6%), with missing information in our analysis, which may have influenced the results. Thirdly, it is important to recognize that a hospital-based study may not be fully applicable to community-based patients, leading to potential selection bias, as our study population was derived from a community hospital in southern Taiwan. This may limit the generalizability of the findings to a broader population. Lastly, the sample size in this study was limited to elderly patients participating in the home-based care program provided by the hospital, which may not fully represent the entire elderly population receiving home-based care in different settings and regions.

## 5. Conclusions

In the present study, we identified that elderly male patients receiving home-based care with an NG tube were at a higher risk for CAP. The challenge for healthcare providers is how to prevent CAP in this particular group of patients. Although the use of PPIs was borderline inversely associated with CAP risk, further prospective and large-scale studies are warranted to validate these findings. Additionally, a higher number of admission days in the last month of life and death in the hospital were found to be associated with end-of-life care for this group. To enhance the quality of end-of-life care for elderly patients with advanced illnesses who receive home-based care and are hospitalized due to CAP, we recommend the implementation of hospice and palliative care specifically tailored to this population. These findings can help healthcare providers and policymakers take proactive measures to improve the care and outcomes for this high-risk group of patients.

## Figures and Tables

**Figure 1 healthcare-12-00443-f001:**
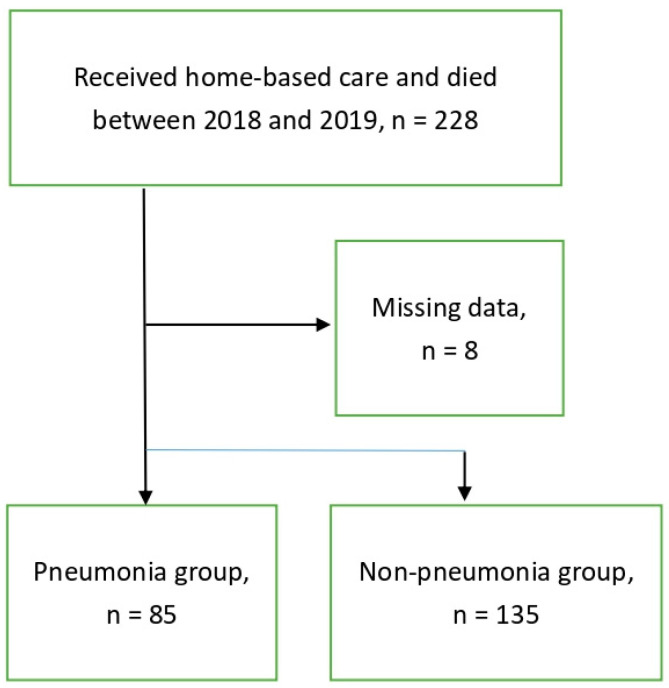
Study design of the current study.

**Table 1 healthcare-12-00443-t001:** Demographic data of the elderly patients.

Variables	Total	Non-P Group	P Group	*p* Value
	(220)	(135)	(85)	
Gender				0.004 ^a^
Female	111 (50.5%)	79 (58.5%)	32 (37.6%)	
Male	109 (49.5%)	56 (41.5%)	53 (62.4%)	
Age	82.0 ± 11.1	81.8 ± 11.4	82.3 ± 10.7	0.846 ^b^
Primary diagnosis				
Cancer	71 (32.2%)	52 (38.5%)	19 (22.4%)	0.017 ^a^
Lung	11 (5.0%)	11 (8.1%)	0 (0)	0.004 ^a^
Colon–rectal	19 (8.6%)	12 (8.9%)	7 (8.2%)	1.000 ^a^
Liver	19 (8.6%)	14 (10.4%)	5 (5.9%)	0.327 ^a^
Others	22 (10.0%)	15 (11.1%)	7 (8.2%)	0.645 ^a^
COPD	13 (5.9%)	7 (5.2%)	6 (7.1%)	0.570 ^a^
Dementia	52 (23.6%)	37 (27.4%)	15 (17.6%)	0.106 ^a^
Parkinsonism	8 (3.6%)	1 (0.7%)	7 (8.2%.)	0.006 ^a^
Stroke	54 (24.5%)	28 (20.7%)	26 (30.6%)	0.109 ^a^
ESRD	12 (5.5%)	5 (3.7%)	7 (8.2%)	0.221 ^a^
CHF	10 (4.5%)	5 (3.7%)	5 (5.9%)	0.514 ^a^
Clinical symptoms/signs				
Unconsciousness	104 (47.3%)	64 (47.4%)	40 (47.1%)	1 ^a^
Fever	49 (22.3%)	29 (21.5%)	20 (23.5%)	0.594 ^a^
Dyspnea	40 (18.2%)	23 (17.0%)	17 (20.0%)	0.594 ^a^
Nausea/vomiting	16 (7.3%)	13 (9.7%)	3 (3.5%)	0.112 ^a^
Respiratory rate, time/min	19.0 ± 5.1	18.9 ± 6.2	18.9 ± 2.75	0.218 ^b^
SBP, mmHg	126.7 ± 20.5	126.6 ± 20.9	126.9 ± 19.8	0.932 ^c^
DBP, mmHg	71.6 ± 12.5	73.2 ± 12.5	69.0 ± 12.1	0.015 ^c^
Heart rate, beat/min	84.5 ± 17.9	84.8 ± 18.3	84.1 ± 17.4	0.814 ^c^
NG tube	146 (66.4%)	74 (54.8%)	72 (84.7%)	<0.001 ^a^
Urinary catheter	101 (45.9%)	63 (46.7%)	38 (44.7%)	0.783 ^a^
Medications				
PPIs	32 (14.5%)	23 (17.0%)	9 (10.6%)	0.239 ^a^
H_2_ blocker	67 (30.5%)	40 (29.6%)	27 (31.8%)	0.765 ^a^
Gastroprokinetic agents *	53 (24.1%)	37 (27.4%)	16 (18.8%)	0.195 ^a^
Days of hospitalization **	8.4 ± 9.2	5.1 ± 7.6	13.6 ± 9.2	<0.001 ^c^
DNR	165 (75.0%)	95 (70.4%)	70 (82.4%)	0.968 ^a^
ICU	12 (5.5%)	7 (5.2%)	5 (5.9%)	1 ^a^
Hospice palliative care	81 (36.8%)	53 (39.3%)	28 (32.9%)	0.390 ^a^
Death in the hospital	147 (66.8%)	75 (55.6%)	72 (84.7%)	<0.001 ^a^

We divided the participants into two groups: elderly patients with advanced illnesses who received home-based care and experienced hospitalization due to CAP (the P group) and those without CAP (the non-P group). Abbreviation: CHF, congestive heart failure; COPD, chronic obstructive pulmonary disease; DNR, Do Not Resuscitate; DSP, diastolic blood pressure; ED, emergency department; ESRD, end stage of renal disease; ICU, intensive care unit; NG, nasogastric; PPIs, proton pump inhibitors; SBP, systolic blood pressure. * Gastroprokinetic agents included Mosapride, Primperan, and Domperidone. ** in the last month of life. ^a^ Fisher’s exact test. ^b^ Wilcoxon rank-sum test. ^c^ two-sample t test.

**Table 2 healthcare-12-00443-t002:** Factors for the community-acquired pneumonia by univariate logistic regression.

Covariates	OR	Estimate	S.E.	Z Value	*p* Value
Male vs. female	2.33 (1.34–4.08)	0.85	0.28	2.99	0.003
Age	1.00 (0.98–1.03)	0.004	0.01	0.28	0.777
Primary diagnosis					
Cancer					
Colon–rectal	0.92 (0.35–2.44)	−0.08	0.50	−0.17	0.867
Liver	0.54 (0.19–1.56)	−0.62	0.54	−1.14	0.255
Others	0.71 (0.28–1.84)	−0.33	0.48	−0.69	0.490
COPD	1.39 (0.45–4.28)	0.33	0.57	0.52	0.567
Dementia	0.57 (0.29–1.11)	−0.57	0.34	−1.65	0.099
Parkinsonism	12.1 (1.45–99.6)	2.49	1.08	2.31	0.021
Stroke	1.68 (0.90–3.13)	0.52	0.32	1.64	0.100
ESRD	2.33 (0.72–7.60)	0.85	0.60	1.41	0.160
CHF	1.63 (0.46–5.79)	0.49	0.65	0.75	0.454
Clinical symptoms/signs					
Unconsciousness	0.99 (0.57–1.70)	−0.01	0.28	−0.05	0.960
Fever	1.12 (0.59–2.15)	0.12	0.33	0.36	0.722
Dyspnea	1.22 (0.61–2.44)	0.20	0.35	0.55	0.579
Nausea/vomiting	0.34 (0.09–1.23)	−1.08	0.66	−1.64	0.101
Respiratory rate, time/min	1.00 (0.95–1.05)	−0.00	0.03	−0.05	0.963
SBP, mmHg	1.00 (0.99–1.01)	0.00	0.01	0.09	0.932
DBP, mmHg	0.97 (0.95–0.99)	−0.03	0.01	−2.40	0.016
Heart rate, beat/min	1.00 (0.98–1.01)	−0.00	0.01	−0.28	0.781
NG tube	4.57 (2.31–9.02)	1.52	0.35	4.37	<0.001
Urinary catheter	0.92 (0.54–1.59)	−0.08	0.28	−0.28	0.776
Medications					
PPIs	0.58 (0.25–1.31)	−0.55	0.42	−1.31	0.190
H_2_ blocker	1.11 (0.61–1.99)	0.10	0.30	0.34	0.738
Gastroprokinetic agents *	0.61 (0.32–1.91)	−0.49	0.34	−1.44	0.149
Days of hospitalization **	1.12 (1.09–1.16)	0.11	0.02	6.15	<0.001
DNR	1.96 (1.01–3.84)	0.68	0.34	1.98	0.048
ICU	1.14 (0.35–3.72)	0.13	0.60	0.22	0.825
Hospice palliative care	0.76 (0.43–1.34)	−0.27	0.29	−0.95	0.345
Death in the hospital	4.43 (2.24–8.76)	1.49	0.35	4.28	<0.001

We divided the participants into two groups: elderly patients with advanced illnesses who received home-based care and experienced hospitalization due to CAP (the P group) and those without CAP (the non-P group). Abbreviation: CHF, congestive heart failure; COPD, chronic obstructive pulmonary disease; DNR, Do Not Resuscitate; DSP, diastolic blood pressure; ED, emergency department; ESRD, end stage of renal disease; ICU, intensive care unit; NG, nasogastric; PPIs, proton pump inhibitors; SBP, systolic blood pressure. * Gastroprokinetic agents included Mosapride, Primperan, and Domperidone. ** in the last month of life.

**Table 3 healthcare-12-00443-t003:** The significant factors for community-acquired pneumonia by logistic regression.

	OR (95% C.I.)	Estimate	S.E.	Z Value	*p* Value
Male vs. female	4.10 (1.95–8.60)	1.41	0.38	3.73	<0.001
NG tube	8.85 (3.64–21.56)	2.18	0.46	4.80	<0.001
PPIs	0.37 (0.13–1.02)	−1.00	0.52	−1.92	0.0546
Admission days *	1.13 (1.08–1.18)	0.12	0.02	5.39	<0.001
Died in hospital	3.59 (1.51–8.55)	1.28	0.44	2.89	0.004
intercept	0.01	−4.61	0.69	−6.68	<0.001

Abbreviation: NG, nasogastric; PPIs, proton pump inhibitors. * in the last month of life.

## Data Availability

The datasets generated during and/or analyzed during the current study are not publicly available, but are available from the corresponding author on reasonable request.

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
