# Peer review of "Factors Associated with Hospitalized Community-Acquired Pneumonia among Elderly Patients Receiving Home-Based Care"

_healthcare, 2024, doi:10.3390/healthcare12040443_

Round 1

Reviewer 1 Report

Comments and Suggestions for Authors

Thank you for the opportunity to read this manuscript. This study aims to identify factors associated with CAP in patients received home-based care, as well as quality indicators of end-of-life care in those who experience CAP. 

1. [62-63] Please clarify which kind of tube this refers to - I presume feeding tubes, but this is not immediately clear upon reading. 

2. [111-113] Were patients categorized by the conditions listed, or are these simply examples of the primary diagnoses documented in charts? Perhaps move up what is written in lines 123-127 to make this clearer.

3. [141-145] Could you clarify the evidence surrounding acid-suppressives and risk of CAP?

4. [151-153] Please define how receipt of hospice palliative care consultation was defined - these services vary significantly in terms of what is provided and who provides it (e.g. RN consult, MD consult, hospice (typically for end of life) vs palliative care (symptom control) as differentiated in the US, etc.). This would be helpful for readers not familiar with the context in Taiwan. 

5. Figure 1 and 201-203: Consider using the word "died" or "deceased" instead of passed away, which is a euphemism not all may be familiar with. 

6. Table 1: How was conscious disturbance defined? Is this referring to delirium? This would be a more familiar term for this reviewer. 

7. [244-245] Please include the Hosmer-Lemeshow test and Nagelkerke R-squared value within the methods section to explain what you are using these for - those not familiar with these tests will struggle to interpret the significance. 

8. I'm struggling to understand why admission days and death in hospital were used as predictors for CAP - these are rather outcomes that would be associated with CAP. Accordingly, I would think there would be two multivariate analyses: 1. CAP, 2. Predictors of poor quality EOL care. 

9. How were admissions days defined for logistic regression? I assume this was split into a categorical variable?

10. Consider linking the discussion around NG tube placement more strongly to your study findings. In my mind, this is one of the most important aspects of this study - enteral feeding tubes may not improve quality of life at EOL (based upon your use of indicators), and it may additionally increase risk of adverse events, such as hospitalization. 

11. Consider shortening the discussion with less review of the existing literature - this reviewer feels that some of the important conclusions of this study are lost within this larger discussion of other publications. 

Author Response

Dear Reviewer, 

Please review the attached word file for our sincere reply. Thank you for your kind review and suggestions.

Reviewer 2 Report

Comments and Suggestions for Authors

Introduction has a too long extension. There are several ideas repeated as lines 44-46 and 72-75

Methods: in the definition of variables (line 129), CAP was defined by both the 3 criteria or just one of them. If the 3, even the diagnosis, were settled to pneumonia, if a patient did not present fever, he was dropped out of the series?

What are the eligibility criteria for home-based care in Taiwan? (line 157)

The exclusion criteria are not the opposite of the inclusion criteria. If you include patients above 65 years old, all others are not excluded from the population: they really are not in the population. If you define the “use of the NG tube and urinary catheter was defined as their timing and duration being for more than 1 week before the diagnosis of CAP”, why did you exclude all those who had NG time less than 1 week?: you are excluding all the patients who were not intubated. However 146/220 patients had NG tube(?)

One of the criteria for inclusion was the death between 2018-2019, correct?

The inclusion criteria should be more clear to understand exactly who are this patients.

Elders (>65 years old)

Had been hospitalised at least 1 day during last month (?)

Death during 2018-19 (or the presence of advanced illness?)

Received home based care (>1 month)

I do not understand what the basic profile of the patients in this study is. Home-based or hospitalised? You took the hospitalized patients to describe those who had home care, or the home care patients to study those who were hospitalised?

Results:

Table 1 should present cancer patients as a whole and then split by organ The lung cancer has 0 patients in P-group: I don’t understand what inferential analysis is the p value of 0.004.

Under the assumption you stated on methods, the PPI presents no effect on pneumonia (CI= 0.13-1.02; p>0.05).

The presence of NG tube must be clarified in the methods.

Discussion

I agree with the risk of prolonged NG tube in elderly. However, your results do not allow to conclude about this since seven days are not prolonged utilization!

This is a case series based on real world patients. There are several limitations of using clinical data retrospectively for research, that should be pointed out in the discussion.

In your conclusion, you state that male and NG intubation are characteristics associated with bad prognosis, and PPI is slightly better. What does this impact on clinical practice? Should I extubate all patients? Should I prescribe more PPI? Especially in males? A short paragraph about it would be well come.

Comments on the Quality of English Language

Language present some minor correction to make

Author Response

(The authors gave the same response as above.)

Round 2

Reviewer 1 Report

Comments and Suggestions for Authors

Thank you for these revisions and clarifications. 

Author Response

Dear Reviewer,

Please review the attached PDF file for our sincere reply. Thank you for your kind review and suggestions.

Reviewer 2 Report

Comments and Suggestions for Authors

Thank you for your efforts to improve your article. My questions about the methods were fully answered and I think it is pretty clear now.

There is just two questions that I must call for your attention.
* You insist that PPI had a borderline effect, as stated in the results and conclusion. However, that is now wholly true. Although the OR is 0.37, the CI crosses the unit and that makes the results not significant, making any other interpretation impossible.

* In Table 1, you use qui-square and fish exact test. It should be clear what test is used for each analysis. I think it would be interesting to join all the patients with cancer for analysis. 71/135 in the non-P group and 19/85 in the P-group (p<0.001). And in a clinical point of view that would make good sense: we think in frailty first in patients with cancer and just then in which cancer they have.

Author Response

Dear Reviewer,

We have revised the manuscript according to your suggestions.
Please review the attached PDF file for our sincere reply.
Thank you for your kind review and suggestions.
